# Defragmenting the 6LoWPAN Fragmentation Landscape: A Performance Evaluation

**DOI:** 10.3390/s21051711

**Published:** 2021-03-02

**Authors:** Amaury Bruniaux, Remous-Aris Koutsiamanis, Georgios Z. Papadopoulos, Nicolas Montavont

**Affiliations:** 1IMT Atlantique, IRISA, 35000 Rennes, France; nicolas.montavont@imt-atlantique.fr; 2IMT Atlantique, STACK (Inria/LS2N), 44000 Nantes, France; remous-aris.koutsiamanis@imt-atlantique.fr

**Keywords:** Internet of Things (IoT), industrial IoT, 6LoWPAN, RFC 4944, fragmentation, fragment forwarding, Forward Error Correction (FEC), network coding

## Abstract

The emergence of the Internet of Things (IoT) has made wireless connectivity ubiquitous and necessary. Extending the IoT to the Industrial Internet of Things (IIoT) places significant demands in terms of reliability on wireless connectivity. The Institute of Electrical and Electronics Engineers (IEEE) Std 802.15.4-2015 standard was designed in response to these demands, and the IPv6 over Low power Wireless Personal Area Networks (6LoWPAN) adaptation layer was introduced to address (among other issues) its payload size limitations by performing packet compression and fragmentation. However, the standardised method does not cope well with low link-quality situations and, thus, we present the state-of-the-art Forward Error Correction (FEC) methods and introduce our own contribution, Network Coding FEC (NCFEC), to improve performance in these situations. We present and analyse the existing methods as well as our own theoretically, and we then implement them and perform an experimental evaluation using the 6TiSCH simulator. The simulation results demonstrate that when high reliability is required and only low quality links are available, NCFEC performs best, with a trade-off between additional network and computational overhead. In situations where the link quality can be guaranteed to be higher, simpler solutions also start to be feasible, but with reduced adaptation flexibility.

## 1. Introduction

As more and more constrained wireless devices are globally connected through the Internet Protocol version 6 (IPv6) [1], a new paradigm called the Internet of Things (IoT) [2] has emerged. Its applications include smart cities, healthcare, power management and Industry 4.0.

The Industrial Internet of Things (IIoT) aims at increasing productivity and efficiency by using IoT devices in order to provide real time monitoring and control, and therefore enabling the automation of production chains. Since losses of data packets could endanger the operation of the production chains, industrial automation networks often require several nines of packet delivery reliability and low latency [3]. Therefore, networking protocols dedicated for industrial networks have to ensure a sufficient Quality of Service (QoS), especially because low-power wireless communications are lossy by nature.

The Institute of Electrical and Electronics Engineers (IEEE) Std 802.15.4-2015 standard was published in 2016, and its Time Slotted Channel Hopping (TSCH) aims at fulfilling such requirements by organising the communications of multi-hop networks with scheduling based on time and frequency. Thus, for each transmission and reception in the network, there is a dedicated cell composed of a timeslot and a radio offset, that is translated into radio channel, to avoid potential collisions between simultaneous communications. However, this link-layer protocol is not adapted to tackle with the IPv6 protocol. Indeed, IPv6 requires the link-layer to be able to transmit packets of at least 1280 bytes while IEEE Std 802.15.4-2015 has a Maximum Transmission Unit (MTU) of 127 bytes. Therefore, an adaptation layer providing fragmentation between the network layer and the link layer, is required to be employed.

The Internet Engineering Task Force (IETF) standardisation organisation defined the IPv6 over Low power Wireless Personal Area Networks (6LoWPAN) in the RFC 4944 standard. It specifies the IPv6 packet compression and fragmentation mechanisms. Moreover, it proposes two methods to route fragments, the Route-Over Routing (ROR) and the Mesh-Under Routing (MUR). All the fragmentation schemes presented in this article are based on ROR scheme because it enables to employ routing protocols such as the IPv6 Routing Protocol for Low-Power and Lossy Networks (RPL), the de facto routing protocol for industrial use-cases.

The standard fragmentation—that will be referenced as **RFC 4944 Fragment Forwarding (RFC 4944 FF)**—allows successfully transmitting IPv6 packets along the mesh networks. However, the potential loss of any fragment makes it impossible to reassemble the original IPv6 packet. This is especially an issue for industrial networks that require several nines of end-to-end reliability and bounded latency. Moreover, networks now require transmitting larger data packets, e.g., configuration data for over-the-air updates or even multimedia data from sensors. In order to tackle this issue, research efforts have focused on the Forward Error Correction (FEC) technique that enables the recovery of dropped fragments by preemptively transmitting additional redundant information over the network. FEC aims at avoiding retransmissions of whole packets, which improves end-to-end reliability and reduces latency.

In this article, we present the standard fragmentation schemes as well as FEC schemes from the literature that improve the reliability of transmissions of fragmented packets by allowing fragment recovery. We also propose modifications to these schemes and specify a new fragmentation method.

The contributions of this article are:We present the state-of-the-art of different FEC fragmentation schemes and propose enhancements that are compatible with the standardised solutions.We propose the Network Coding FEC (NCFEC) new fragmentation scheme that uses end-to-end network coding, a new fragment structure and adaptation to the quality of the network radio links in order to achieve end-to-end network reliability.We evaluate the performance of these fragmentation techniques based on simulations performed on the 6TiSCH simulator [4].

Then in the rest of the article, Section 2 describes the standard 6LoWPAN fragmentation and Section 3 exposes the ongoing standardization activities at the IETF. Subsequently, in Section 4, we describe three new approaches based on the FEC technique to improve the end-to-end reliability in multi-hop networks. We continue with Section 5, which provides the theoretical analysis of the performance of the fragmentation schemes, while Section 6 evaluates them with the simulation results. Finally, Section 7 concludes the article and provides future directions.

## 2. Technical Background: RFC 4944

### 2.1. Overview

RFC 4944 [5] is the main standard for the 6LoWPAN protocol that specifies the adaptation layer for IPv6 packet transmission over IEEE Std 802.15.4-2015 links. This article focuses on the fragmentation aspect of this standard. The IPv6 protocol requires that layer 2 links handle a packet size, i.e., MTU, of at least of 1280 bytes. However, the IEEE Std 802.15.4-2015 comes with an MTU of 127 bytes, which means that an adaptation layer is required in order to make it compatible with IPv6. 6LoWPAN addresses the issue of the MTU and allows handling packets of size up to 2048 bytes by performing compression and splitting the IPv6 packets into several fragments.

The IPv6 and User Datagram Protocol (UDP) headers, which have an original size of 40 bytes and 8 bytes, respectively, are compressed according to RFC 6282 [6]. If the resulting packet is still larger than 127 bytes, it requires fragmentation: it is split into several link fragments of up to 127 bytes, each containing the Medium Access Control (MAC) header, the fragment header and the payload of the IPv6 packet. The first fragment also contains the compresssed IPv6 and UDP headers.

The fragment header, shown in Figure 1, is 4 bytes long for the first fragment, 5 bytes long for the remaining ones, and contains the four following fields:Dispatch type (5 bits): 11000 for a first fragment and 11100 for all subsequent fragments.Datagram size (11 bits): field that contains the information about size of the IPv6 packet before fragmentation in bytes.Datagram tag (16 bits): field that is used to identify all the fragments from a single IPv6 packet. It increments for each new fragmented packet and its maximum value is 65,535.Datagram offset (8 bits): field that represents the offset of the fragment from the beginning of the packet. The first fragment has an offset of 0 and omits this field. The subsequent fragments contain this field and the offset increment is 8 bytes.

There are two ways these fragments can be routed toward their destination: mesh-under and route-over. MUR is performed over the multi-hop network and takes place at the adaptation layer, which means the IPv6 packets do not need to be reassembled at each hop because the link-layer address is contained in each fragment, and thus each fragment is routed individually. ROR is performed at the network layer and uses the IPv6 destination address to route packets. In RFC 4944 FF, the first fragment is the only one containing this address, thus the need to reassemble packets at each hop with ROR in order to access this information and route whole packets, which increases end-to-end latency. Even though MUR offers shorter delays with the standard fragmentation, ROR is preferred because it allows the use of IPv6 addresses and, thus, enables the use of routing protocols such as RPL, the de facto routing protocol for industrial use-cases. Therefore, all fragmentation schemes and examples presented in this article are based on the ROR scheme.

With ROR, when a node receives the first fragment of a packet for which it has no allocated reassembly buffer, it creates a buffer of size equal to the datagram size field in the fragment header. It also starts a timer, at the end of which the fragments of the packet will be discarded if any of them are missing. When the subsequent fragments belonging to a packet having an entry in the buffer arrive, they are placed in the buffer. When the buffer is full, the packet can be reconstructed. After a node successfully reconstructs a packet, if itself is not the destination of the packet, the packet must be forwarded.

The FEC schemes presented in this article tackle the reassembly issue of ROR either by using the Virtual Reassembly Buffers (VRBs) proposed by [7], temporarily saving the information needed to route fragments, or by adding a partial IPv6 header to each fragment.

The example illustrated in Figure 2 depicts the transmission of an IPv6 packet sent by a leaf node to the root node and because it is larger than the MAC payload limit, it requires to be fragmented into four fragments. All fragments are successfully transmitted to the next hop, which is a relay node that creates a reassembly buffer into which it stores the fragments before transmitting them to the root node. All transmissions are successful and therefore the root node can reassemble the packet from the fragments it stored in its reassembly buffer.

### 2.2. Issues with RFC 4944 FF

In [8], a number of issues that negatively affect the performance of RFC 4944 FF are presented. Firstly, the nodes have limited memory and can only contain a limited number of reassembly buffers. Therefore, when a fragment from a new packet is received and the node does not have enough space to create a new buffer it has to drop a packet. Another issue caused by the reassembly at each hop is that it entails end-to-end delays as each node has to wait for the reception of all the fragments of a packet before forwarding them. Additionally, several source nodes can use the same datagram tag for their own IPv6 packets. This entails confusion if a node receives fragments from different flows sharing the same datagram tag and causes forwarding issues. These three issues are addressed by the Minimal Fragment Forwarding (MFF) scheme using a VRB at each hop to forward fragments.

Finally, in order to reassemble a packet, every fragment has to be received by the destination. If any fragment is lost at any hop in the path the packet will not be received by the destination. This is illustrated by the example depicted in Figure 3, in which an IPv6 packet is sent by a leaf node to the root and has to be fragmented into four fragments. However, the third fragment will not be successfully transmitted from the relay node to the root node. Therefore, after the reassembly timer of the root node expires, it must discard all the fragments and the whole transmission fails. Addressing this issue is the initial motivation for the use of FEC techniques.

## 3. Ongoing Standardisation Efforts: Minimal Fragment Forwarding (MFF)

The work on MFF [7] is a soon-to-be standard that addresses several of the issues of the RFC 4944 FF that were previously presented. MFF forwards the fragments without performing the reassembly and the re-fragmentation processes at each hop. To do so, it uses a VRB to route all fragments by storing a minimal routing information that is carried only in the first fragment.

### 3.1. Overview

The fragmentation process is performed by the source node, while the reassembly is performed by the destination node according to the RFC 4944, as presented in the previous section. The updates to the RFC 4944 standard is in regard to the fragment forwarding scheme that is performed in the intermediate nodes. Indeed, when an intermediate node receives a fragment with a datagram tag for which it has no VRB entry:It checks if the datagram offset is 0. If it is not, it means that the first fragment is lost and the final destination will not be able to reassemble the packet. This fragment and all the subsequent from the same IPv6 packet will be dropped immediately by the intermediate node.If the datagram offset is 0, it creates a VRB entry containing:
The received datagram tag.A new datagram tag that will be created and used to forward the fragment.The MAC address of the previous hop.The MAC address of the next hop (decided at reception of the first fragment and used to forward the subsequent fragments).A timer used to discard the buffer after a timeout.It forwards the fragment to the next hop

Storing the link-layer address of the previous hop allows differentiating fragments with the same datagram tag sent by different nodes. For instance, in Figure 4, node G receives fragments with datagram tag = 3 from two different nodes but will be able to differentiate the incoming fragments from the two nodes because it stored their link-layer address in the corresponding VRB entry. It will also change the datagram tags of the outgoing fragments so that node H is able to differentiate them, therefore solving the confusion issue mentioned in the previous section.

On the other hand, when an intermediate node receives a fragment of an IPv6 packet for which it already has a VRB entry, it forwards the fragment immediately to the next hop without waiting for the whole packet to arrive. This is illustrated by Figure 5, in which an IPv6 packet is sent by the leaf node to the root node and is fragmented into four fragments. Once the relay node receives the first fragment, it creates a new entry in the VRB table, where it stores the necessary information that will allow it to immediately forward the following fragments. The destination node, typically the root node, once it has successfully received the four fragments, can reassemble the original IPv6 packet.

### 3.2. Performance Evaluation of the 6LoWPAN FF and the MFF

To further illustrate the reassembly issues with RFC 4944 FF that are solved with MFF, we performed a performance evaluation using the 6TiSCH simulator [4]. We simulated a multi-hop network with the bottleneck at node 1 as illustrated in Figure 6, where the leaf nodes 5 and 9 are transmitting IPv6 data packets to the root node every 40 s with a link quality at every hop of 0.85, a TSCH slotframe size of 101 and each node given 15 random transmission cells per slotframe. As it can be seen in Figure 7, the MFF scheme considerably improves both the latency and the end-to-end network reliability compared to the 6LoWPAN FF. This observed difference in terms of network reliability is due to the fragment handling at node 1, i.e., at the bottleneck. In this series of simulations, the nodes have enough memory only for one reassembly buffer. Therefore, under the 6LowPAN FF scheme, when node 1 receives a fragment from node 5 while it has a buffer containing fragments from node 9, it has to drop one of them. On the other hand, with the MFF mechanism, fragments are forwarded without being buffered and, thus, this queue overflow issue does not occur.

### 3.3. Issues with MFF Scheme

Although MFF allows for faster fragment forwarding along the multi-hop network, and addresses several of RFC 4944 FF issues such as latency, datagram tag confusion and memory usage, other issues remain. Firstly, intermediate nodes still use buffers. Despite the fact that the memory required for a VRB is much lower than that for a reassembly buffer, the number of VRB entries is still finite. Therefore, if there are more packets to forward simultaneously than the number of VRB entries, packets will still be dropped. Secondly, no fragment loss can be recovered from and losses can entail additional traffic. Indeed, if any fragment is lost, the process of fragment forwarding will continue, unless the first fragment is the one lost, in which case the next hop will drop all the fragments. However, the destination will still not be able to reassemble the packet and, thus, there will be unnecessary traffic in the network. This is illustrated in Figure 8, where an IPv6 packet larger than the MTU is transmitted from the leaf node to the root node and is fragmented into four fragments. The second fragment will not be successfully transmitted by the leaf node to the relay node, while all the other fragments are successfully transmitted along the network, especially the first one that enables the new registration in the VRB table. In such a scenario, the fragments transmitted by nodes after the loss of the second fragment introduce unnecessary traffic because under no circumstance will the root node will be able to recover the missing fragment. Additionally, only after the timer expires will the root discard the received fragments, leading to buffer memory waste.

Finally, as pointed out by [9], since the first fragment is the only one which contains the IPv6 header, it is required to be received before the subsequent fragments to enable a routing decision. This makes it impossible to use multi-path routing without repeating the first fragment on every path.

## 4. Forward Error Correction (FEC)

FEC is a technique that can be used to ensure reliable transmission, without needing the destination node to ask for retransmission of missing fragments, by adding redundant information to sent packets. Used in satellite communications, mobile networks and Low Power Wide Area Networks (LPWANs) [10], FEC mechanisms often use encoding algorithms enabling the destination to recover missing elements. 6LoWPANs are deployed on lossy radio links, that can cause loss of part of the fragments of a packet, which in the case of RFC 4944 FF or MFF entails the failure of the reception of the whole packet. Adding FEC to 6LoWPAN addresses this issue by making missing fragments recoverable. This article compares three FEC methods for 6LoWPANs that introduce redundant information by repeating fragments or adding encoded fragments sharing the information of several original fragments. XORFEC adds a unique encoded fragment, Repetition FEC (RFEC) adds identical repetition of all fragments and NCFEC encodes all fragments of sent packets.

Although FEC mechanisms significantly improve network reliability in lossy environments, they also entail additional costs. Indeed, the additional fragments require energy and bandwidth to be sent, and the increase in traffic can ultimately lead to overflow in the transmission queues of relay nodes. Therefore, FEC schemes should be implemented carefully in networks with a high traffic load and where the bandwidth is limited. In order to address the traffic increase, in [11], the authors propose deactivating FEC when the network presents low loss probability, while we propose a mechanism that lowers the number of fragments when it is not detrimental to the QoS. FEC can also enable the reception of packets that would otherwise be lost but this option comes with additional delay since the fragments used in the recovery process are sent after the original fragments. Finally, the FEC schemes that use encoding require the nodes to be able to perform the encoding and decoding operations with costs of additional computations. This performance evaluation illustrates the cost of FEC by comparing the latency and the traffic of FEC schemes and standard fragmentation.

### 4.1. XORFEC

#### 4.1.1. Overview

XORFEC is a FEC mechanism that uses the Exclusive OR (XOR) operator (⊕) to generate an additional fragment for a fragmented IPv6 packet. This additional fragment contains redundancy and enables achieving higher reliability. Indeed, this additional fragment is sent after the original fragments of the packet and allows the destination node to recover from the loss of one original fragment. As a result, the consequence of losing a single fragment is no longer the loss of the whole packet.

XORFEC extends Network Coding - Mesh Under Routing (NC-MUR) [12]—which uses MUR, meaning that each fragment is routed based on the link-local address of its MAC header—in order to be compatible with the MFF that uses ROR without reassembly at each hop. Indeed XORFEC fragments use the forwarding scheme of MFF, including the additional fragment thanks to an adapted 6LoWPAN header.

#### 4.1.2. The XOR Operator

XOR is a logical operator that is used in encoding mechanisms in order to mix together the information from several packets and decode the encoded packets when they are received. XOR is a binary operator and, when applied to packets of several bits, is applied bitwise. The property of XOR that is used in XORFEC and that allows recovering fragments is that applying XOR on the result of a first XOR and one of the inputs of this first XOR gives the second input:(1)$$\begin{array}{c} B=A⊕(A⊕B) \end{array}$$(2)$$\begin{array}{c} A=B⊕(A⊕B). \end{array}$$

Indeed, if a packet is fragmented into two fragments *A* and *B*, the additional fragment *C* generated by the source node will be:(3)C=A⊕B.

Then, if the destination receives *A* and *C* but does not receive *B*, it can recover *B* by applying the XOR operator to the fragments it received. This operation can be generalised to packets with a larger number of fragments: with *m* original fragments ($M_{1},\ldots ,M_{m}$), the additional fragment *N* will be:(4)$$N=M_{1}⊕\ldots ⊕M_{m}.$$

If the destination node receives all the transmitted fragments except for $M_{i}$, it can recover the latter by applying the operator to the fragments received:(5)$$M_{i}=\left(M_{1}⊕\ldots ⊕M_{i-1}⊕M_{i+1}⊕\ldots ⊕M_{m}\right)⊕N.$$

However this recovery is only possible if no more than one fragment is missing, i.e., the loss tolerance of XORFEC is one missing fragment.

#### 4.1.3. Operation

When a 6LoWPAN node requires transmitting an IPv6 packet of a size larger than the layer 2 MTU (127 bytes), it will fragment it into *n* fragments where only the first one will contain the compressed IPv6 header. Therefore, the loss of this first fragment cannot be recovered. Then, an additional fragment is generated by applying the XOR operator to the *n* original fragments. The 6LoWPAN header is added to each fragment as defined in the RFC 4944 standard.

For the additional fragment, the datagram offset will be the continuation of other offsets. For instance, if the original IPv6 packet to transmit is of size 450 bytes, and the 6LoWPAN payload size is 90 bytes, the original fragments will have offsets 0, 90, 180, 270, 360 and the additional fragment will have an offset of 450. This makes XORFEC compatible with MFF, as nodes not using XORFEC can ignore the additional fragments that have a datagram offset higher than the datagram size. In such a case, the packets will be treated as regular MFF packets without the option of fragment recovery. After the fragmentation, the source node transmits the *n* original fragments followed by the additional fragment.

Regarding the forwarding mechanism of relay nodes, it follows similar principles as in MFF: using VRBs to forward fragments without waiting for reassembly at each hop. When the destination node receives the first fragment of a new packet, it creates a reassembly buffer of the size given by the datagram size present in the header. When any *n* fragments of the IPv6 packet have successfully arrived at the destination, the original packet can be reassembled. If the additional fragment is among the first *n* received fragments, an original fragment has been lost. In that case the missing fragment is recovered with an XOR operation on all received fragments, and after this operation the packet is reassembled. Figure 9 illustrates this process where an IPv6 packet is transmitted from the leaf node toward the root node. The packet is first fragmented into four original fragments and the additional fragment is generated by applying the XOR operator to these four fragments. When it receives the first fragment, the relay node creates the VRB, based on the 1st fragment that contains the IPv6 header, in order to forward the fragments. The third fragment gets lost between the leaf node and the relay node but is recovered by the root node thanks to the additional fragment. If one more fragment had been lost, the recovery would not have been possible, as the additional fragment only contains enough redundant information to recover one missing fragment.

### 4.2. RFEC

#### 4.2.1. Overview

We introduce RFEC, which is a FEC-based mechanism and extended version of the algorithm presented in [11] that introduces redundancy by identically repeating each fragment of a packet, without any encoding. In [11], the authors chose to activate FEC only when the PDR falls below a pre-defined threshold so that it does not generate unnecessary traffic. When FEC is activated, fragments are repeated by the source node three seconds after the original fragments are sent. This delay is added in order to avoid the loss of both original and retransmitted fragments in a short term perturbation of the radio quality.

We extend the algorithm presented in [11] with RFEC that also repeats each fragment but does not add delay and the copies are transmitted consecutively to their originals. Moreover, VRBs are used to forward packets at intermediary nodes in order to avoid the reassembly process at each hop.

#### 4.2.2. Operation

Because RFEC uses VRBs to forward the fragments without reassembly, the loss of the first fragment would compromise the creation of the VRB and the other original fragments could not be routed. Therefore, removing the delay and sending each copied fragment immediately after its original could improve the reliability: in case of loss of the original first fragment, its copy can be used to create the buffers that will route the other original fragments. This is the case in the scenario Figure 10 depicts where an IPv6 packet is sent by the leaf node to the root node of a two-hop network. It is first fragmented into four original fragments, and each fragment is transmitted twice consecutively. Even though it does not receive the first copy of the first fragment, the relay node receives the second copy that enables it to create a VRB. This VRB allows the relay node to forward every fragment of the packet. The root node receives one copy of the first, second and fourth fragments and two copies of the third fragment. It drops the second copy of the third fragment and is able to reassemble the packet.

### 4.3. NCFEC

NCFEC is yet another FEC-based mechanism that uses network coding to generate a set of encoded fragments so that any fragment can be lost as long as a sufficient number of fragments are received by the destination node. Network coding is a technology first introduced in 2000 by [13] and originally aiming at optimizing the bandwidth usage of multi-source networks by making nodes performencoding and decoding operations [14]. Network coding also has applications in the fragmentation domain where it is used to improve the reliability of transmissions, as illustrated in [15]. In NCFEC, the number of encoded fragments is also higher than the number of original non-encoded fragments to add more reliability. The employed encoding is based on Reliable IPv6 Packet Delivery Scheme (RIPDS) [16], where the reassembly, fragmentation and encoding are performed at each hop. Moreover in RIPDS, the number of encoded fragments is set at each link *i* as $\left\lceil m/{PDR}_{i}\right\rceil$ where *m* is the number of original fragments and ${PDR}_{i}$ is the PDR at link *i*.

The main features of NCFEC are:Fragmentation and encoding are only performed by the source node, while the decoding and reassembly are only performed by the destination node. This decreases both latency, and the usage of computational resources.Each fragment contains a compressed IPv6 header that enables forwarding without using a VRB, which allows forwarding subsequent fragments in the case of a loss of the first fragment. Thus, each fragment can be routed individually, and multi-path routing can be applied.An end-to-end PDR estimation mechanism using RPL’s DODAG Information Object (DIO) packets is performed by the source node in order to evaluate the number of encoded fragments it needs to generate.

#### 4.3.1. Fragment Structure

In this article, we introduce a new dispatch type, i.e., encoded fragment, with its header shown in Figure 11. There are several dispatch patterns reserved for future use in RFC 4944, and 11 011 xxx is one of them. The MAC payload of a packet can either be 102 bytes without security or 81 bytes with security [17]. After adding the new NCFEC header, the available payload for encoded fragments is 93 bytes without security or 72 bytes with security, as shown in Figure 12 that illustrate the NCFEC fragmentation process. In the example depicted, the three original fragments are encoded into five fragments.

#### 4.3.2. Original Fragments Generation

The addition of new fields reduces the available space for the fragment payload. Let $n_{p}$ be the new payload, and $p_{s}$ be the IPv6 packet size. The first fragmentation step is to split the IPv6 packet (header included) into the original fragments. The number of original fragments $N_{originalfragments}$ is calculated as: (6)$$N_{originalfragments}=\left\lceil p_{s}÷n_{p}.\right\rceil$$

#### 4.3.3. Galois Field GF(2,8)

Before presenting how the fragments are encoded, GF(2,8) needs to be introduced. This is the Galois finite field with $2^{8}$ elements. The elements of GF(2,8) can be represented by several forms: polynomial, binary or integer. For example, 42 has a binary form: 00101010 and can also be represented by the polynomial $x^{6}+x^{4}+x^{2}$. In this field, the operations are not the usual ones. The addition is the XOR operator. The multiplication is the polynomial multiplication modulo an irreducible polynomial of degree 8.

#### 4.3.4. Encoding

Once the original fragments are generated, the encoded fragments need to be created. There are more encoded fragments than there are original fragments, and several ways to determine the number of encoded fragments are detailed later in this section. Let *m* be the number of original fragments and *M* the number of encoded fragments. For each encoded fragment *i*, i∈{1,2,…,M}, we allocate *m* coefficients $x_{i,k}$:(7)$$x_{i,k}=i^{k-1},k\in \left\{1,2,\ldots ,m\right\}.$$

Each coefficient is an element from the Galois field GF(2,8). The fragments are encoded one word at a time, with a word being a group of 8 bits. The words are then all juxtaposed to create the encoded fragment. Let $p_{k,l}$ be the *l*th word of the *k*th original fragment, and $q_{i,l}$ the *l*th word of the *i*th encoded fragment. These words are also elements of GF(2,8). Each word of the encoded fragments is obtained as follows:(8)$$q_{i,l}=\sum_{k=1}^{m}x_{i,k}p_{k,l},$$
where the addition and the multiplication are those defined for GF(2,8).

#### 4.3.5. NCFEC Operation

The source node sends all the fragments to the destination node. When an intermediate node receives a fragment, it reads the destination address, chooses the next hop accordingly and forwards the fragment. After *m* different fragments from the same packet have arrived, the destination node can reassemble the packet. When a fragment arrives at its destination node:The node checks if it has a reassembly buffer entry corresponding to the pair (source address, datagram tag) of the fragment. If it does not, it creates a reassembly buffer of size *m* (known by reading the datagram size field in the fragment header), containing the source address and the datagram tag and starts a timer.The node adds the fragment to the corresponding buffer entry.If *m* fragments are in the buffer, the node starts the reassembly procedure.

If the timer of a buffer entry is elapsed, all the fragments are discarded.

#### 4.3.6. Determining the Number of Additional Fragments

In [16], the encoding was repeated at each hop, and the number of packets chosen was $\left\lceil m/{PDR}_{i}\right\rceil$ with ${PDR}_{i}$ being the estimated PDR of the link. The authors chose this value because this way an average of *m* fragments would be received for each packet, *m* being the lowest number of received fragments allowing reassembly.

However, this value does not ensure the desired high reliability as every time the number of fragments received is below this average value the packet cannot be reassembled.

In order to achieve high reliability levels, it is necessary to increase the information redundancy and, therefore, increase the number of fragments. In this article, we opted to use a reliable way to determine the lowest number of encoded fragments sufficient to achieve an end-to-end PDR target, based on PDR estimation.

#### 4.3.7. Estimation of PDR

Each node is capable of estimating the end-to-end PDR for one fragment, which is used to derive the packet end-to-end PDR. Indeed, in RPL, DIO packets have the option of containing the Directed Acyclic Graph (DAG) metric container. We employed the Expected Transmission Count (ETX) Reliability Object metric presented in [18] with the recorded option. The ETX is the number of transmissions a node expects to perform to successfully transmit a fragment to its current parent. It is computed by counting both the MAC packets sent and the MAC Acknowledgments (ACKs) received. It is also an estimation of the inverse of the link PDR between the sending node and its parent.
(9)$$ETX=TX÷ACK=1÷PDR_{link}.$$

Given the ETX for each hop, and knowing the number of maximum MAC retransmissions, a node is able to compute the end-to-end PDR for one fragment and for a whole IPv6 packet. The details for these estimations are presented in Section 5.

In the example scenario depicted in Figure 13, a packet destined to the root node is fragmented into four fragments by the leaf node. These four original fragments are used to generate six encoded fragments encapsulated in the NCFEC packet format and sent to the relay node. Upon reception of a fragment, the relay node routes it using the destination address present in its header. Therefore, even though it does not receive the first encoded fragment the relay node is able to forward all the fragments, which would not be possible with the other schemes using VRBs or reassembly. When any four fragments are received, the root node decodes the encoded fragments in order to obtain the four original fragments and is then able to reassemble the packet.

#### 4.3.8. Adaptive Number of Fragments Selection

NCFEC takes a target performance (minimal end-to-end IPv6 packet PDR) as a network parameter and aims at achieving this target while using as few resources as possible. In other words, its goal is to determine the lowest number of encoded fragments sufficient to achieve its end-to-end PDR target. It performs this computation before sending each IPv6 packet, so that if it detects a variation in the link qualities it will adapt accordingly by changing the number of encoded fragments generated.

The source node has access to the end-to-end PDR for one fragment, therefore it can compute the end-to-end PDR for a whole IPv6 packet for a different set of parameters. Here, the parameter that changes is the number of encoded fragments. The nodes perform a computation to test if the number of fragments considered is sufficient and increase the number until the estimated reliability is above the target parameter. If the number of fragments exceeds a threshold, the computation is stopped and the threshold value is chosen, in order to limit the traffic overhead. Without this limitation the number of fragments could reach extremely high numbers in low radio quality scenarios.

We included this PDR estimation in order to adapt the network configuration to the conditions of its environment. More specifically, ensuring a high level of network reliability when the link quality degrades, or limiting the unnecessary traffic increase when it is not required. However, this estimation is based on the assumption that the time scale at which the link quality changes is slower than the time scale of the estimation. The estimation is recomputed each time the source node receives an ETX update of the nodes in the path towards its destination. The frequency of these updates depends on the computation of the ETX by each node and the delivery of DIO packets. In case this frequency is too low, the estimation will not be in sync with the current link quality and will lead to a sub-optimal choice of the number of fragments. A low PDR estimation leads to unnecessary traffic, while a high estimation leads to a lower number of fragments than is required in order to achieve the expected network reliability. Basing the number of encoded fragments on this estimation is better suited to networks with relatively stable link quality.

## 5. Mathematical Analysis

### 5.1. Theoretical PDR

In order to compare the performance of the fragmentation methods detailed in the previous section, we determined the theoretical end-to-end PDR for each algorithm. Note that in this section we call PDR the probability of reception of fragments and packets as it is the equivalent of the expected PDR that would be measured. End-to-end PDR here means PDR for one application packet over several hops with the following parameters:$PDR_{i}$ is the probability of reception of one fragment at link i$r_{i}$ is the maximum number of total Medium Access Control (MAC) layer transmissions at link i*n* is the number of original fragments*H* is the total number of hops


**End-to-end fragment PDR**


In order to compute the end-to-end PDR for a whole packet, it is necessary to compute the end-to-end PDR for a single fragment. This intermediate step will be used in all following computations. To obtain this end-to-end PDR for a single fragment, we first need to compute the probability of reception of one fragment over one hop. The probability of reception of one fragment over the link *i* after all possible MAC retransmissions can be expressed as:(10)$$PDR_{f/h}\left(i\right)=1-{\left(1-PDR_{i}\right)}^{r_{i}}.$$

Then the probability of reception of one fragment end-to-end is:(11)$$PDR_{f/e}=\prod_{i=1}^{H}PDR_{i,f/h}.$$


**RFC 4944 FF and MFF schemes**


For RFC 4944 FF, the issues described in Section 3.2 (i.e., buffer overflow and datagram tag confusion) are not taken into account for these computations. Therefore, the actual expected PDR is lower, especially if the packet transmission frequency is high.

To allow reassembly of an IPv6 packet, all fragments have to be received by the destination node. Hence, the probability of end-to-end reception of the packet is:(12)$$PDR_{MinimalFragment}=PDR_{f/e}^{n}.$$


**XORFEC**


To allow the reassembly of the packet, the first fragment and at least n−1 fragments among the *n* have to be received. Thus, the probability of end-to-end reception of the packet is:(13)$$PDR_{XORFEC}=PDR_{f/e}P\left\{R\geq n-1\right\},$$
with R being the random variable counting the number of non-first fragments received. It follows the binomial law:(14)P{R=k}=nkPDRf/ek(1−PDRf/e)n−k.


**NCFec**


Let $N_{o}$ be the number of original fragments and $N_{e}$ the number of encoded fragments. To allow the reassembly of the packet, any $N_{o}$ encoded fragments among the $N_{e}$ sent fraggments have to be received. Therefore, the probability of end-to-end reception of the packet is:(15)$$PDR_{NCFec}=P\left\{N_{r}\geq N_{o}\right\},$$
with $N_{r}$ being the random variable counting the number of encoded fragments received. It follows the binomial law:(16)P{Nr=k}=NekPDRf/ek(1−PDRf/e)Ne−k.


**RFEC**


Considering that the RFEC scheme can be applied with and without additional latency before the transmission, we computed the packet end-to-end PDR for both. Under the hypothesis that the link quality does not change over the duration of the transmissions of the fragments, removing the delay and sending the repeated packets just after their original copies improves the reliability because the copy of the first fragment can be used to create a VRB that will allow to forward all next fragments.

With delay, if the original first fragment is not received, the rest of the original fragments cannot be used as there is no VRB.
(17)$$PDR_{RFec-delay}=PDR_{f/e}{\left(1-{\left(1-PDR_{f/e}\right)}^{2}\right)}^{n-1}+\left(1-PDR_{f/e}\right)PDR_{f/e}^{n}.$$Without delay:
(18)$$PDR_{RFec-nodelay}=\left(2PDR_{f/e}-PDR_{f/e}^{2}\right){\left(1-{\left(1-PDR_{f/e}\right)}^{2}\right)}^{n-1}.$$

### 5.2. Illustration of Theoretical Performance

Figure 14a shows three-dimensional plots of the computed theoretical PDR for different packet sizes, link qualities and fragmentation methods, while Figure 14b shows the number of fragments that should be generated by the source node for each packet. Figure 14a shows that FEC mechanisms achieve higher reliability than the IETF standard fragmentation. In particular, NCFEC allows achieving reliability above 0.99 as a target PDR with link quality as low as 0.5 while with MFF and even the other FEC mechanisms performance drops. However, this reliability improvement comes with an increase of network traffic as more fragments need to be sent in order to ensure that a sufficient number of fragents is received.

## 6. Performance Evaluation

### 6.1. Simulation Setup

We performed extensive simulations to evaluate the performance presented by the use of MFF, XORFEC, RFEC and NCFEC with simulations performed in the IPv6 over the TSCH mode of IEEE 802.15.4e (6TiSCH) simulator [4]. In this discrete-event simulator written in Python, both the RFC 4944 FF and the MFF fragment forwarding standards are implemented on top of the 6TiSCH stack. The simulations were executed on the Grid’5000 platform [19]. We have performed simulations based on a linear topology of 10 nodes, as depicted in Figure 15. Node 9 is the sole source of data traffic, while the root node is the destination. Regarding the schedule at the MAC layer, we employed a random fixed schedule, where at the beginning of each run, each node is given 20 random TSCH cells to transmit to its parent if it has one, and 20 reception cells to receive packets from the child node, if it has one. The probability of correct transmission of each link is the same, and the number of maximum retransmissions at the MAC layer is 3, which makes four total possible transmission opportunities. The simulation parameters used are detailed in Table 1.

To evaluate the performance of each algorithm, we have employed the following three performance indicators:End-to-end network reliability: we measure the end-to-end packet PDR.End-to-end latency: we measure the end-to-end delay for the received packets.Traffic: we measure the number of fragments that are generated per original IPv6 packet. For NCFEC, this value can be changed throughout the simulation as the estimated radio quality changes and it depends on the targeted PDR, i.e., 0.99.

The performance evaluation results with a link quality of 0.65 are presented in Figure 16, and with a link quality of 0.85 in Figure 17.

### 6.2. Simulation Results

With all fragmentation methods when the packet size is smaller than the MAC MTU, the source node does not proceed with fragmentation and, thus, it transmits the packet directly. As expected, the FEC based mechanisms achieve better end-to-end network reliability than the standard, and trade off traffic and latency. Indeed this reliability comes at the cost of sending additional fragments forwarded throughout the network and these fragments are sent—and thus received—after the original fragments. The increase in reliability happens as soon as fragmentation is required. For instance, with a link quality of 0.65, MFF packets of two fragments are received with a PDRof 77%, whereas with the lightest FEC scheme, XORFEC, the reliability increases up to 87%. Moreover, the reliability increase is even more significant when the packet sizes are larger, in terms of bytes. This increase in reliability justifies the relevance of FEC in networks where the additional traffic and delay is tolerable.

In these simulations, NCFEC was able to achieve its target PDR of 0.99 without exceeding its maximal number of fragments allowed and therefore achieves its goal of very high reliability. However, when the link quality is low, this reliability is achieved at the cost of a large traffic increase. The FEC based mechanisms have a significant impact when the transmission conditions are poor, i.e., low link quality and larger packets. In these conditions, the effect of the sole additional fragment that XORFEC adds is notable, it allows adding 14% of reliability with two original fragments and 32% with 10 original fragments as shown in Figure 16c. Furthermore, the difference in reliability is smaller between XORFEC, NCFEC and RFEC when the link quality is high, i.e., XORFEC being a more simple and less costly mechanism is more adapted to this situation.

This simulation campaign demonstrates the cost of FEC fragmentation schemes, as they show both a latency and a traffic increase. XORFEC will always add one fragment to each packet compared to MFF, while NCFEC adds four to six fragments with a link quality of 0.65 and always two fragments with a link quality of 0.85. That represents an increase of traffic of 60% to 200% with a link quality of 0.65 and of 20% to 100% with a link quality of 0.85 in order to achieve 99% end-to-end network reliability. The highest relative increase of traffic happens for packets of two original fragments and NCFEC is more efficient operating with larger packets. Therefore, if the reliability requirements allow it, a network could be designed to use XORFEC for small packets, and NCFEC for larger packets in order to limit the traffic increase. RFEC always doubles the amount of traffic but has the benefit of not performing encoding which makes it better suited for networks where nodes have very limited computing power.

## 7. Conclusions

In this article, we have demonstrated the relevance of FEC to improve the end-to-end network reliability in multi-hop 6LoWPAN networks and presented three FEC mechanisms. XORFEC adds a sole additional fragment obtained via a simple operation, RFEC repeats all fragments identically and NCFEC applies network coding to make any fragment loss recoverable. We carried out a performance analysis of these FEC methods compared against the standard 6LoWPAN fragmentation. FEC allows better reliability than the standard fragmentation at the cost of higher traffic load. However, the three FEC based mechanisms come with different performance and costs which makes them more adapted to worse network conditions. Indeed, NCFEC always achieves the highest reliability but requires higher computing resources. Then, RFEC is a simple and reliable algorithm but entails the highest increase in traffic and in latency. Finally, XORFEC comes with minimum impact and could still be sufficient in high link quality scenarios, but is not performant in lower-quality link situations. The simulations performed and the results obtained for different link qualities can guide the selection of an appropriate FEC mechanism, given the link qualities and the computational resources available in the target application.

For future work, we expect to evaluate the different fragmentation methods under more variable conditions: with topologies where the link quality changes throughout the simulation. Moreover, we would like to go deeper into the evaluation of these algorithms by performing real world experiments, for instance using the FIT IoT-LAB platform [20].

## 8. Materials and Methods

The simulations of Section 6 have been performed using the 6TiSCH simulator [4].

## Figures and Tables

**Figure 1 sensors-21-01711-f001:**
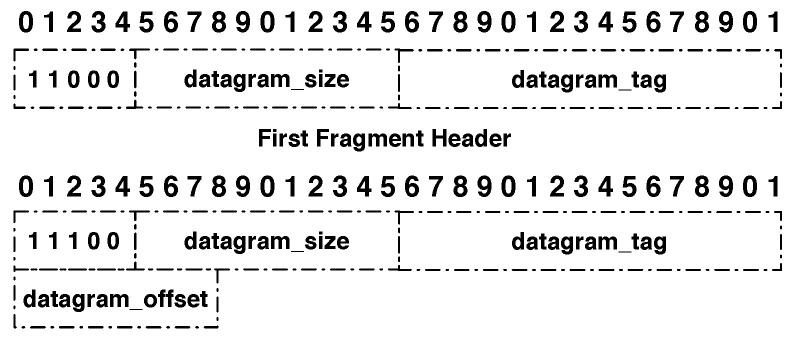
6LoWPAN header defined by RFC 4944.

**Figure 2 sensors-21-01711-f002:**
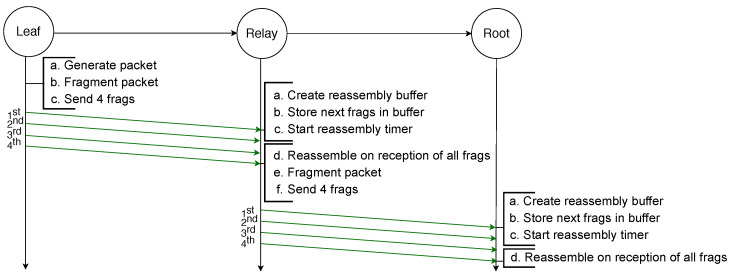
Successful transmission of a fragmented packet with standard 6LoWPAN over two hops.

**Figure 3 sensors-21-01711-f003:**
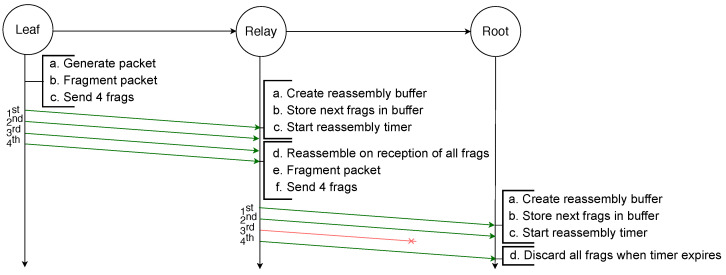
Failed transmission of a fragmented packet with RFC 4944 FF over two hops: the loss of one fragment makes the reassembly of the packet impossible.

**Figure 4 sensors-21-01711-f004:**
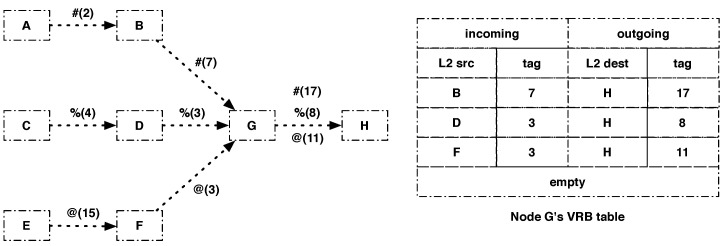
VRB table of node G: #(2), %(4) and @(15) are fragments of the packets denoted #, %, and @ coming from nodes A, C and E, respectively, with datagram tag configured to 2, 4 and 15, respectively.

**Figure 5 sensors-21-01711-f005:**
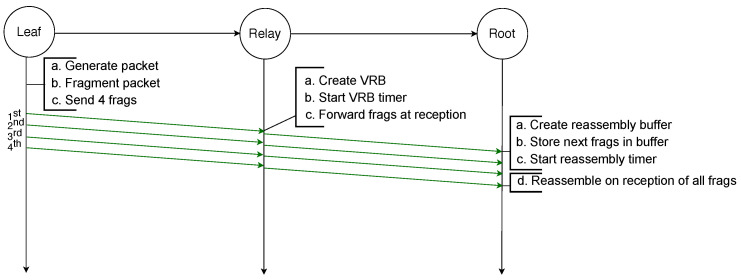
Successful transmission of a fragmented packet over two hops with MFF.

**Figure 6 sensors-21-01711-f006:**
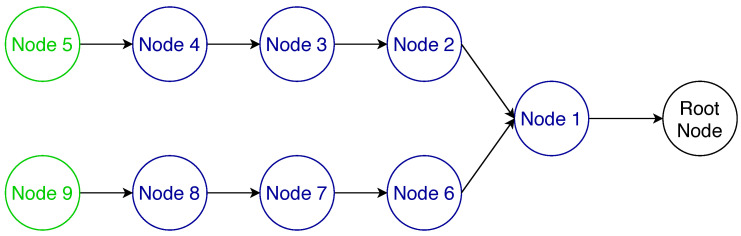
Bottleneck topology.

**Figure 7 sensors-21-01711-f007:**
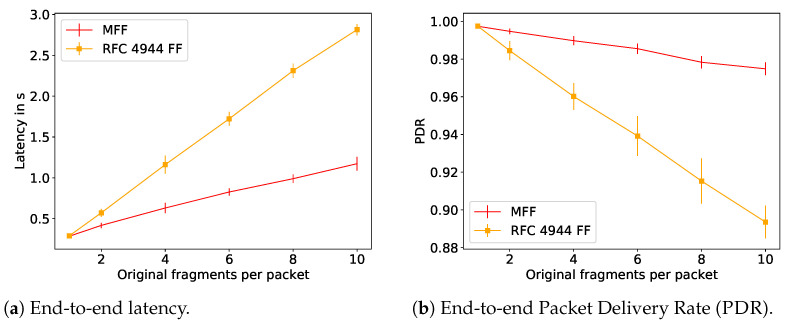
Performance evaluation of the schemes proposed by the IETF.

**Figure 8 sensors-21-01711-f008:**
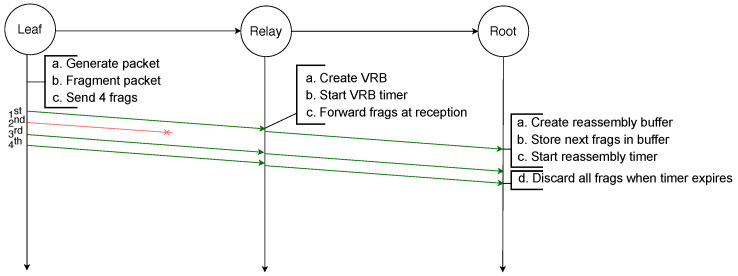
Unsuccessful transmission of a fragmented packet over two hops with MFF: the loss of the second fragment is not recoverable.

**Figure 9 sensors-21-01711-f009:**
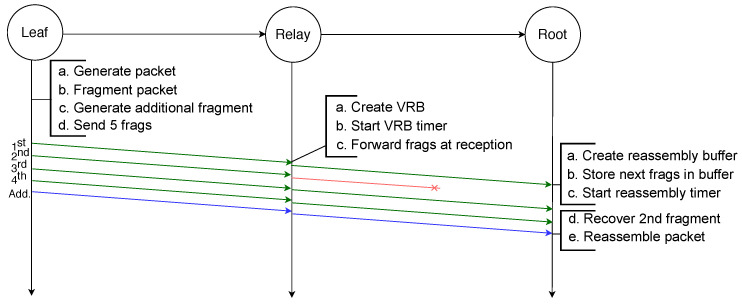
Successful transmission of a fragmented packet with XORFEC: even though one fragment is lost it is recovered thanks to the additional fragment.

**Figure 10 sensors-21-01711-f010:**
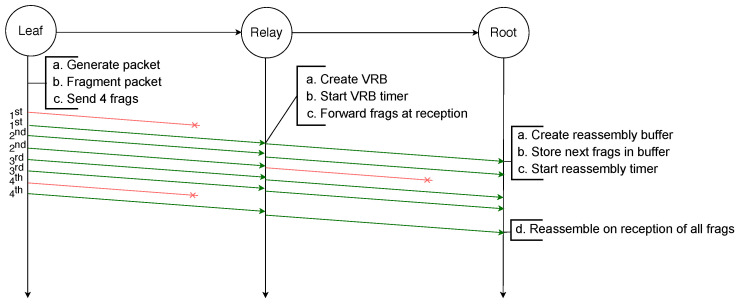
Successful transmission of a packet with RFEC.

**Figure 11 sensors-21-01711-f011:**
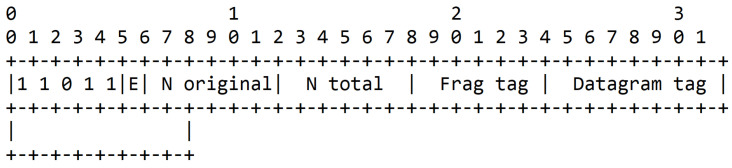
Structure of the 6LoWPAN fragment header with NCFEC.

**Figure 12 sensors-21-01711-f012:**
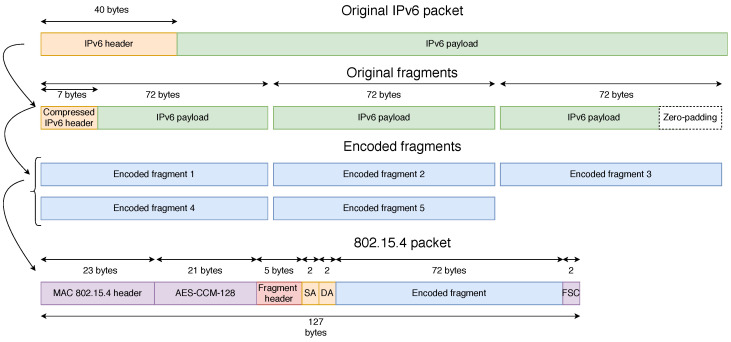
Structure of the MAC frame with NCFEC. SA and DA stand for Source Address and Destination Address of the IPv6 packet.

**Figure 13 sensors-21-01711-f013:**
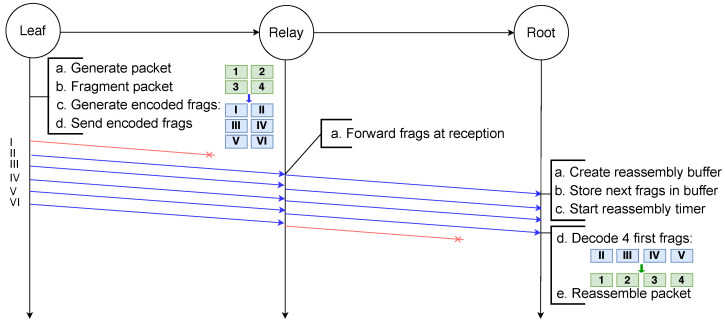
Successful transmission of a fragmented packet with NCFec despite the loss of several fragments.

**Figure 14 sensors-21-01711-f014:**
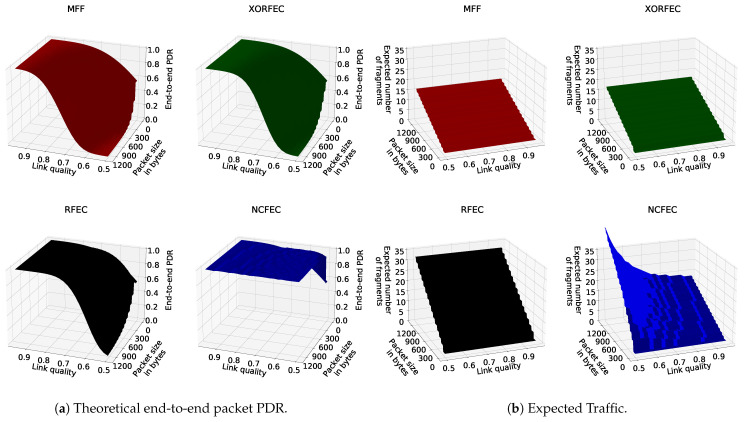
Theoretical performance.

**Figure 15 sensors-21-01711-f015:**
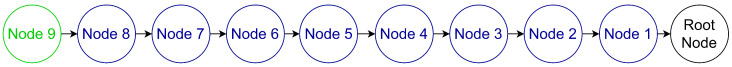
Linear topology.

**Figure 16 sensors-21-01711-f016:**
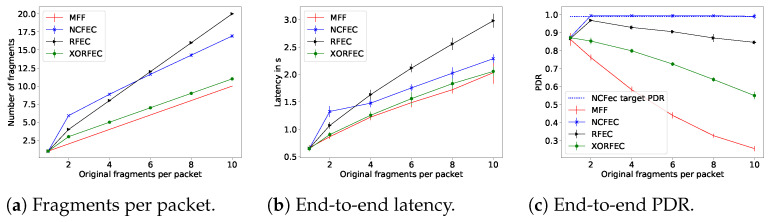
Simulation results for linear topology with link quality of 0.65.

**Figure 17 sensors-21-01711-f017:**
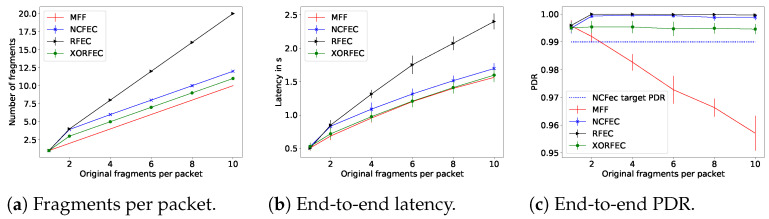
Simulation results for linear topology with link quality of 0.85.

**Table 1 sensors-21-01711-t001:** Simulation parameters.

Parameter	Settings
TSCH slotframe length	101 slots
Slot duration	10 ms
Link reliability	Between 0.65 and 0.95
Maximum number of retries	3
Packet interval	Uniform in [54 s, 66 s]
Number of fragments per packet	Between 1 and 10
Number of simulation runs	100
Duration of each simulation run	1000 s
NCFEC target PDR	0.99
NCFEC maximum redundancy factor	3

## Data Availability

Data not available.

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
