# Peer review of "Defragmenting the 6LoWPAN Fragmentation Landscape: A Performance Evaluation"

_sensors, 2021, doi:10.3390/s21051711_

Round 1

Reviewer 1 Report

The paper is well written and provides interesting results. An improvement would be to provide a discussion on possible dependencies related to the end-to-end PDR estimation mechanism, continuously changing link quality and non-evenly distributed loss patterns. This is now vaguely mentioned as part of the indicated future work in the conclusion section, but is (as far as I can see). not mentioned earlier in the paper at all. I suggest to include an explanation on what dependencies there are between the NCFEC end-to-end PDR estimation and other possible relevant conditions e.g. end-to-end delay, loss characteristics and time-scales at which link qualities changes.

Author Response

Thank you for your comments, please see the attachment.

Reviewer 2 Report

This paper addresses the 6LoWPAN fragmentation scheme and in particular how it does not perform well in poor link quality scenarios. The authors review existing FEC schemes, propose a new one (Network Coding FEC or NCFEC) and demonstrate its performance via simulation. Generally, the paper is well written and provide an approachable background to the subject and research.  

Section 2
Although this section is understandable and presents a generally good introduction to some 6LoWPAN fragmentation issues, it could be clearer. For example, at line 109, the text discusses the need for re-assembly of packets (without any qualification), but then at line 117, it is stated that re-assembly is not required for MUR if the fragments are to be forwarded. I think the text needs to clearly delineate between MUR and ROR to prevent any confusion, possibly using separate sub-sections. I am assuming Figures 2 and 3 are for ROR rather than MUR, but the text is not explicit in this regard.
  Another potential issue is that the terms "packet" and "datagram" are both used. I believe their use is interchangeable in this text, even though "packet" sometimes refers to layer 3 (IP) and "datagram" to layer 4 (UDP). Either these terms need to be clearly defined and then used precisely in the correct context, or the authors should choose just one of them to prevent any confusion.  

Section 3
 This section is generally well explained.   Check the caption for Figure 5 (what does "draft fragmentation" mean?). Figure 7(b): PDR should be defined on first use here (I realise it is defined in the Acronyms section, but every acronym should still be defined on first use in the main text)  

Section 4
 I think there does need to be some discussion about the drawbacks of employing FEC as well as the advantages. There is no such thing as a free lunch. The additional fragments needed when FEC is employed create additional utilisation and latency, and may result in some routing buffers being overloaded whereas they might not have been if FEC had not been employed. How do the authors justify the use of FEC and how much FEC to apply in this regard? There is some discussion about this subject in Sections 6 and 7, but I think it should be discussed from a design perspective too - what level of design compromise is optimum and why?   In Section 4.2.2, describing the operation of RFEC with the aid of Figure 10, I find it odd that only the source/leaf node repeats fragments. The relay node just seems to relay what it receives, and if it only receives one of the two repeats, it does not itself regenerate the repeat. This inherently makes the second hop less reliable than the first hop. I think there should be some commentary on why all nodes (including relay nodes) do not send repeats.  

Section 5
This section is generally well explained.  

Section 6
I think Figures 16 and 17 should be together so the reader can more easily observe the effect of link quality. T his section is quite short for a journal paper, I would expect more (simulation) results and analysis. Perhaps you could show graphs of latency and PDF versus link quality as the independent variable?

Typos

The Abstract twice refers to "exiting" FEC methods instead of "existing" FEC methods.
Line 74: "The in the rest of the article..." Line 116: "There are two ways these fragment can be routed..." - 'fragment' should be plural. Line 127: "The fragmentation schemes presented in
128 this article tackle this issue either by using Virtual Reassembly Buffer (VRB)s,..." - 'Buffer should be plural and the 's' should be inside the parentheses.
Line 338: "This VRB allows the relay node to forward every arriving of the packet." - replace 'arriving' with 'fragment'? Line 360: "Each fragment contains a compressed IPv6 header that enables forwarding without using a VRBs" - 'VRB' should be singular Figure 11 and 12 have the same title/caption. I think the title of Figure 12 should be "Structure of the MAC frame".

Author Response

(The authors gave the same response as above.)

Reviewer 3 Report

The paper is well written. The authors present a method a new fragmentation method called Network Coding FEC (NCFEC) and propose modifications to other existing schemes. They can be improved related work or discuss other proposals. I suggest the authors to present the overhead to their proposal. For example, what is the time and the computational costs to defragment the packets?

Author Response

(The authors gave the same response as above.)
